# Cost-effectiveness of trastuzumab deruxtecan for previously treated HER2-low advanced breast cancer

Demin Shi[1☯], Xueyan Liang[2☯], Yan Li[2], Lingyuan Chen[iD][3]*

1 Department of Reproductive Medicine, The People's Hospital of Hechi, Hechi, Guangxi, People's Republic of China, 2 Department of Pharmacy, Guangxi Academy of Medical Sciences and the People's Hospital of Guangxi Zhuang Autonomous Region, Nanning, Guangxi, People's Republic of China, 3 Department of Pharmacy, The People's Hospital of Hechi, Hechi, Guangxi, People's Republic of China

☯ These authors contributed equally to this work.
* lingyuanchen@outlook.com

## Abstract

### Objective

The clinical efficacy and safety profile of trastuzumab deruxtecan (T-DXd) have been demonstrated in previously treated patients with human epidermal growth factor receptor 2 (HER2)-low advanced breast cancer (BC). It is, however, necessary to evaluate the value of T-DXd considering both its clinical efficacy and its cost, given that it is high. This study aimed to evaluate the cost-effectiveness of T-DXd versus chemotherapy in patients with previously treated HER2-low advanced BC.

### Methods

We used a partitioned survival model that included three mutually exclusive health states. The patients in the model were identified based on their clinical characteristics and outcomes from the DESTINY-Breast04. Probabilistic and one-way sensitivity analyses were performed to evaluate the model's robustness. Subgroup analyses were also conducted. The measures included costs, life years, quality-adjusted life-years (QALYs), incremental cost-effectiveness ratios (ICERs), incremental net health benefits (INHBs), and incremental net monetary benefits (INMBs).

### Results

The ICERs of T-DXd vs. chemotherapy were \$83,892/QALY, \$82,808/QALY, and \$93,358/QALY in all HER2-low advanced BC patients, HER2-positive (HER2+) advanced BC patients and HER2-negative (HER2-) advanced BC patients, respectively. In one-way sensitivity analysis, the cost of T-DXd and hazard ratio (HR) for progression-free survival (PFS) and overall survival (OS) were also identified as key drivers. If the price of T-DXd decreased to \$17.00/mg, \$17.13/mg, and \$14.07/mg, it would be cost-effective at a willingness to pay (WTP) threshold of \$50,000/QALY in all HER2-low advanced BC patients, HER2+ advanced BC patients and HER2- advanced BC patients, respectively. At a WTP threshold

**Data Availability Statement:** All relevant data are within the paper and its Supporting Information files.

**Funding:** This work was supported by the scientific research and technological development projects of Hechi, Guangxi Province of China (Heke B1824–4). The funders had no role in study design, data collection and analysis, decision to publish, or preparation of the manuscript.

**Competing interests:** The authors have declared that no competing interests exist.

of $100,000/QALY, the probability of T-DXd being cost-effective was 81.10%, 82.27%, and 73.78% compared to chemotherapy for all HER2-low advanced BC patients, HER2+ advanced BC patients and HER2- advanced BC patients, respectively. Most subgroups of patients with HER2+ disease had a cost-effectiveness probability of > 50%.

## Conclusion

From a third-party payer's perspective in the United States, the findings of the cost-effectiveness analysis revealed that, at the current price, T-DXd is a cost-effective alternative to chemotherapy for patients with prior HER2-low advanced BC, at WTP threshold of $100,000/QALY.

## Introduction

Breast cancer (BC) is the most common cancer among women [1]. Early detection, surgery, radiation therapy, and systemic treatment have improved the overall survival (OS) rate of patients diagnosed with stage I-III BC to 70–80% [2], however, approximately 30% of women diagnosed with early-stage BC progress to advanced or metastatic cancer [3]. Poor prognosis coupled with the high incidence of BC has made it the second leading cause of cancer-related mortality in women worldwide after lung cancer, primarily because of the adverse prognosis for most patients with advanced disease [1, 4].

In addition to patient and disease characteristics, the molecular subtypes of BC play an important role in prognosis and therapeutic decision-making. Approximately 20% of all BC cases are associated with overexpression of human epidermal growth factor receptor 2 (HER2) [5]. In the last two decades, the landscape of systemic treatment for advanced BC has changed significantly, with main developments occurring in the treatment of hormone receptor-positive (HR+)/HER2-negative (HER2-) as well as HER2-positive (HER2+) cases [4]. Several types of HER2-targeted therapies have been used in metastatic setting, including trastuzumab-emtansine, trastuzumab, pertuzumab, and lapatinib [6]. About 60% of BC cases with HER2-metastatic expression exhibit low levels of HER2, defined as an immunohistochemical (IHC) score of 1+ or 2+ IHC score combined with a negative in situ hybridization result (ISH) [7, 8]. The term "HER2-low" is used to describe both HR+ and hormone receptor-negative (HR-) BCs that will respond differently to systemic chemotherapy depending on their prognosis [7, 8]. Patients with this subtype have not benefited from existing HER2-directed therapies [9, 10]; therefore, BC with a HER2-low status is currently treated as HER2- (HER2-low and HER2-zero), with patients stratified by hormone receptor status [4, 8, 11]. In general, patients who progress after primary therapy have limited options for targeted treatment, and most commonly receive palliative chemotherapy containing a single agent [4, 8, 11].

Trastuzumab deruxtecan (formerly DS-8201, T-DXd), has been approved to treat metastatic HER2+ BC patients with a cleavable linker containing a humanized anti-HER2 monoclonal antibody linked to topoisomerase I inhibitor [12, 13]. Through its bystander effect, T-DXd can deliver its cytotoxic payload to neighboring tumor cells that are heterogeneously expressing HER2 in addition to targeting HER2-expressing tumor cells with high levels of HER2 [13, 14]. Previously published RCTs have demonstrated promising results in pretreated patients with metastatic HER2-low advanced BC [15, 16]. As reported in the recently published phase 3 RCT DESTINY-Breast04, 52.3% of HER2-low advanced BC patients had an overall response, ranging from 9.0 to 11.3 months of progression-free survival [16]. The results

of these studies suggest that T-DXd is highly effective in treating HER2-low advanced BC patients [16, 17].

To the best of our knowledge, no cost-effectiveness analyses have been conducted comparing T-DXd with chemotherapy for advanced BC. To effectively distribute the limited healthcare resources to clinicians and decision-makers, cost-effectiveness analyses are necessary. Thus, T-DXd was evaluated from the perspective of a third-party payer in the United States (USA) for the treatment of patients with previously treated HER2-low advanced BC patients.

## Materials and methods

### Patients and intervention

This study was conducted following the Consolidated Health Economic Evaluation Reporting Standards (CHEERS) [18]. This study did not require approval from the institutional review board because it used data available in the public domain and open databases instead of individual patient data.

Hypothetical target patients with HER2-low metastatic BC were selected from the DESTINY-Breast04 randomized clinical trial [16]. This study included patients with unresectable or metastatic BC that was HR+ (inferred from local testing that 1% of tumor cell nuclei are immunoreactive for estrogen or progesterone receptors) or HR-. In the DESTINY-Breast04 trial [16], patients in the T-DXd group received 5.4 mg per kilogram every three weeks, whereas those in the physician's choice group received eribulin (51.1%), capecitabine (20.1%), nab-paclitaxel (10.3%), gemcitabine (10.3%), and paclitaxel (8.2%).

### Model structure

A partitioned survival model was used to evaluate the economic impact of progression-free survival (PFS), progressive disease (PD), and death [19–21]. Both treatment arms had a 10-year time horizon and > 98% of the patients died during that period. The cycle length was one week. Based on the results of the DESTINY-Breast04 study, the proportions of target patients with OS and PFS were used [16]. The area under the OS curve was evaluated as a measure of the proportion of patients living with PD, and the difference between the OS and PFS curves was calculated as a measure of the proportion of patients living with PD.

The main outcomes were overall costs, incremental cost-effectiveness ratios (ICERs), quality-adjusted life years (QALYs), life years (LYs), incremental net health benefits (INHB), and incremental net monetary benefits (INMB) were the main outcomes of our study. A willingness to pay (WTP) threshold of $100,000/QALY was established [22]. A discount of 3% per year was applied to all cost and utility outcomes [19, 21].

### Clinical data input

Patients with HER2-low advanced BC in the T-DXd and chemotherapy groups were obtained from the DESTINY-Breast04 trial [16]. According to the algorithm developed by Guyot et al., OS and PFS were extrapolated beyond the trial follow-up period [23]. To extract individual patient data, GetData Graph Digitizer version 2.26 [24] was used to generate Kaplan-Meier survival curves for OS and PFS. We fitted the following parametric survival functions to these data points: exponential, Weibull, gamma, lognormal, Gompertz, log-logistic, and generalized gamma. Subsequently, the Akaike information criterion (AIC) and Bayesian information criterion (BIC) values were used to determine the best-fit parametric model for the reconstructed K-M survival curves. The survival functions and parametric models for T-DXd and chemotherapy treatment are presented in Table 1 and S1 Table provides the goodness-of-fit results

**Table 1. Key model inputs.**

| Parameter | Value (95% CI) | Distribution | Source |
|---|:---:|:---:|:---:|
| **Clinical input** | | | |
| **Scenario 1 All patients** | | | |
| **Survival model for trastuzumab deruxtecan** | | | |
| Log-logistic model for OS[a] | γ = 1.6742, λ = 0.0095 | ND | [16] |
| Lognormal model for PFS[a] | μ = 3.7343, σ = 1.1400 | ND | [16] |
| **Survival model for chemotherapy** | | | |
| Log-logistic model for OS[a] | γ = 1.7826, λ = 0.0136 | ND | [16] |
| Lognormal model for PFS[a] | μ = 3.0558, σ = 1.0395 | ND | [16] |
| **Scenario 2 Hormone receptor-positive cohort** | | | |
| **Survival model for trastuzumab deruxtecan** | | | |
| Log-logistic model for OS[a] | γ = 1.79219, λ = 0.0093 | ND | [16] |
| Lognormal model for PFS[a] | μ = 3.77703, σ = 1.12199 | ND | [16] |
| **Survival model for chemotherapy** | | | |
| Log-logistic model for OS[a] | γ = 1.72949, λ = 0.0125 | ND | [16] |
| Lognormal model for PFS[a] | μ = 3.11192, σ = 1.04968 | ND | [16] |
| **Scenario 3 Hormone receptor-negative cohort** | | | |
| **Survival model for trastuzumab deruxtecan** | | | |
| Log-logistic model for OS[a] | γ = 1.3437, λ = 0.0120 | ND | [16] |
| Lognormal model for PFS[a] | μ = 3.48501, σ = 1.21098 | ND | [16] |
| **Survival model for chemotherapy** | | | |
| Lognormal model for OS[a] | μ = 3.692111, σ = 0.837671 | ND | [16] |
| Lognormal model for PFS[a] | μ = 2.731798, σ = 0.889911 | ND | [16] |
| **Cost input** | | | |
| **Drug costs per 1 mg** | | | |
| Trastuzumab deruxtecan | 29.68 (23.74 to 35.62) | Gamma | [26] |
| Capecitabine | 0.0027 (0.0022 to 0.0033) | Gamma | [29] |
| Eribulin | 1294 (1035 to 1552) | Gamma | [29] |
| Gemcitabine | 0.0199 (0.0159 to 0.0239) | Gamma | [29] |
| Paclitaxel | 0.128 (0.102 to 0.154) | Gamma | [29] |
| Nab-paclitaxel | 14.19 (11.35 to 17.02) | Gamma | [29] |
| Cost of terminal care per patient[b] | 20409 (16327 to 24491) | Gamma | [32] |
| **Disease management and monitoring costs** | | | |
| CT scan of chest (per time) | 135 (58 to 256) | Gamma | [27] |
| Best supportive care (per cycle) | 472 (377 to 566) | Gamma | [28] |
| **Cost of managing AEs (grade ≥ 3)[c]** | | | |
| Trastuzumab Deruxtecan | 4586 (3669 to 5503) | Gamma | [3, 30, 31] |
| Chemotherapy | 5896 (4717 to 7076) | Gamma | [3, 30, 31] |
| **Administration cost** | | | |
| First hour | 159 (130 to 206) | Gamma | [27] |
| Additional hour | 34 (28 to 42) | Gamma | [27] |
| Follow-up cost per events | 251 (223 to 318) | Gamma | [27] |
| **Health utilities** | | | |
| **Disease status utility per year** | | | |
| Stable disease | 0.830 (0.664 to 0.935) | Beta | [3, 34] |
| Disease progression | 0.443 (0.354 to 0.532) | Beta | [3, 35] |
| Death | 0 | NA | |
| **Disutility due to AEs[d]** | | | |

*(Continued)*

**Table 1.** (Continued)

| Parameter | Value (95% CI) | Distribution | Source |
|---|---|---|---|
| Trastuzumab Deruxtecan | 0.027 (0.022 to 0.033) | Beta | [3, 30, 31] |
| Chemotherapy | 0.023 (0.018 to 0.028) | Beta | [3, 30, 31] |
| **Other parameters** | | | |
| Body surface area, m$^2$ | 1.82 (1.44 to 2.16) | Normal | [36] |
| Body weight, kg | 74 (59 to 90) | Normal | [36] |

[a]Only expected values are presented for these survival model parameters

[b]Overall total cost per patient regardless of treatment duration.

[c]Calculated as the average cost of toxic effects using weighted frequencies of grade $\geq$ 3 treatment related adverse events for each treatment arm in the DESTINY-Breast04 trial. Costs of individual toxic effects were derived from the literature and include all care required to manage each toxic effect. References for individual toxic effect costs are summarized in S2 Table in the Supplement.

[d]Calculated as the average disutility of toxic effects using weighted frequencies of grade $\geq$ 3 treatment-related adverse events for each treatment arm in the DESTINY-Breast04 trial. Disutilities of individual toxic effects were derived from the literature. References for individual toxic effect disutilities are summarized in S2 Table in the Supplement.

Abbreviations: CMS, Centers for Medicare & Medicaid Services.

for these two treatments. The OS and PFS K-M curves of T-DXd and chemotherapy in different scenarios were fitted using log-normal and log-logistic models, which are presented in S1 Table and S1 Fig.

## Cost data input

In this study, we evaluated direct medical costs, including the cost of drugs, costs associated with supportive care, administration costs, costs associated with terminal care, costs associated with adverse events (AEs), and costs associated with CT scans (Table 1). The prices of drugs were gathered from public databases [25, 26]. The Medicare Clinical Laboratory Fee Schedule was used to obtain the cost of CT scans [27], supportive care [28], administration fees [27], and follow-up fees [29]. We estimated the cost of treating severe adverse events [3, 30, 31] and terminal care [32] using published costs and applied them as one-time events (Table 1). According to the Medical-Care Inflation data obtained from Tom's Inflation Calculator, all costs have been adjusted to US dollars for 2021 and were inflated to 2021 monetary values [33].

## Health utility inputs

Health utility scores were assigned on a scale ranging from 0 (death) to 1 (perfect health). Since DESTINY-Breast04 does not provide health utility scores for PFS and PD, we used the health utility obtained from published scores [3, 34]. The utilities of PFS and PD associated with advanced BC were 0.830 and 0.443, respectively, which were derived from a cost-effectiveness analysis that included patients with advanced BC [3, 34, 35]. The disutility values related to adverse events were also obtained from the literature (S2 Table) [3, 30, 31].

## Base case analysis

Three scenarios were analyzed (Fig 1), including all HER2-low advanced BC patients (scenario 1), HER2+ advanced BC patients (scenario 2), and HER2- advanced BC patients (scenario 3). Throughout all scenarios, the treatment regimens received by the patients remained the same, and only the survival data differed. Base-case analysis was performed using the ICER to

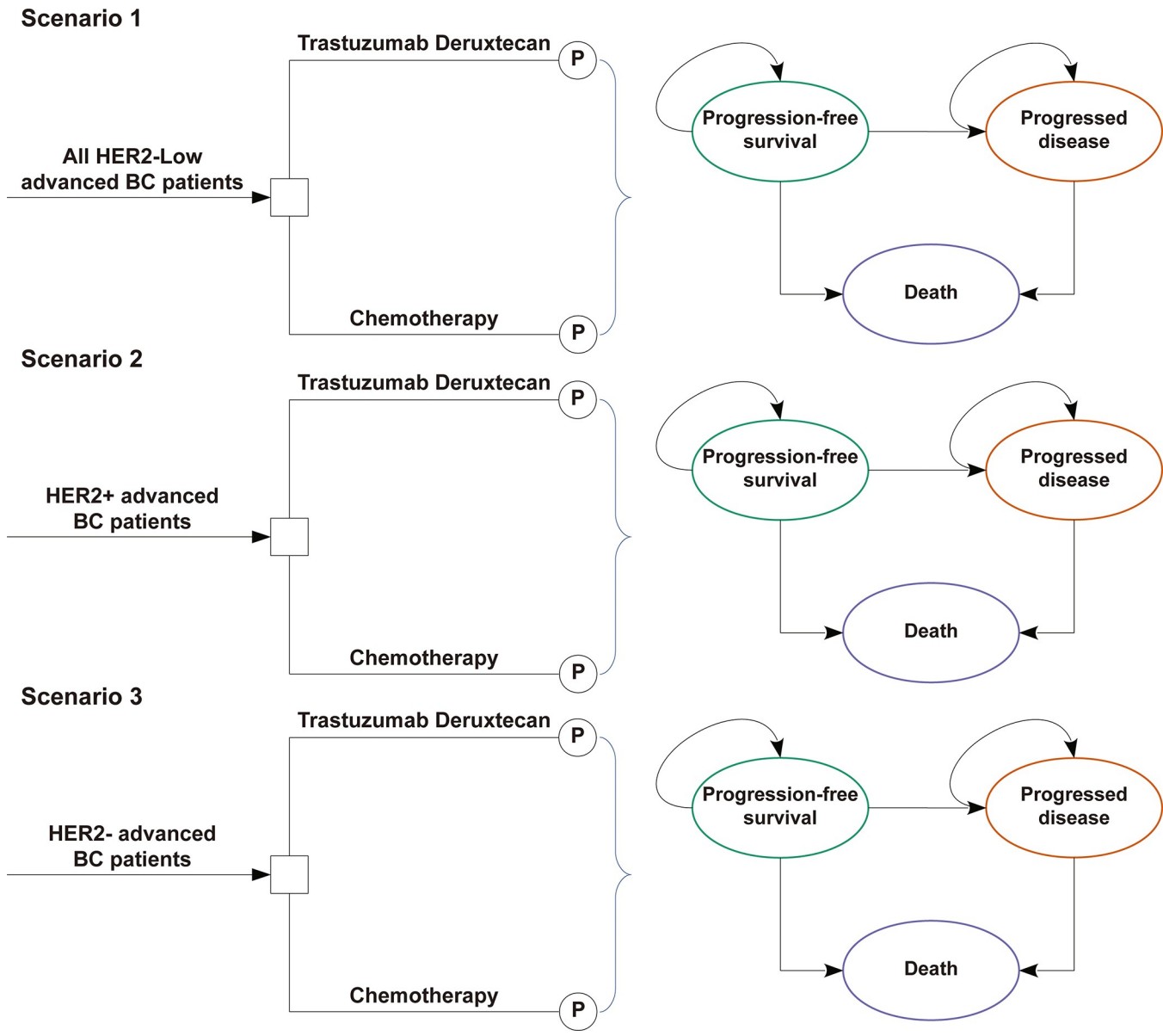

**Fig 1. The partitioned survival model consisting of three discrete health states.** Abbreviations: BC, breast cancer; P, partitioned survival model.

measure additional QALYs. According to published literature [22], the WTP threshold in the United States is $100,000. According to the recommendations, cost-effectiveness was assumed in cases where the ICER was lower than the WTP threshold ($100,000/QALY). To estimate the chemotherapy dosage, we assumed that a typical patient had a body surface area of 1.82 m$^2$ and a weight of 74 kg [36]. Additionally, we calculated the INHB and INMB [19, 21]. The INHB and INMB are calculated as follows: $\text{INHB}(\lambda) = \left( \mu E_{\text{T–DXd}} - \mu E_{Chemotherapy} \right) - \frac{(\mu C_{\text{T–DXd}} - \mu C_{Chemotherapy})}{\lambda} = \Delta E - \frac{\Delta C}{\lambda}$ and $\text{INMB}(\lambda) = \left( \mu E_{\text{T–DXd}} - \mu E_{Chemotherapy} \right) \times \lambda - \left( \mu C_{\text{T–DXd}} - \mu C_{Chemotherapy} \right) = \Delta E \times \lambda - \Delta C,$ where μC and μE were the cost and utility of T-DXd or chemotherapy, respectively, and λ was the WTP threshold.

### Sensitivity analysis

We conducted a one-way sensitivity analysis to identify variables that were significantly sensitive and assessed the model's robustness. One-way sensitivity analyses were conducted according to various variables such as costs and utilities, and the uncertainty was estimated by applying the 95% confidence intervals (CIs) in the literature or assuming that the fundamental parameters would vary by 20% (Table 1). Monte Carlo simulations were utilized for probabilistic sensitivity analysis, which was conducted with 10,000 iterations. In all cases, a suitable distribution was determined based on different parameters. For cost parameters, proportion and probability, and baseline characteristics, gamma, beta, and normal distributions were assigned, respectively. Subsequently, a cost-effectiveness acceptability curve was constructed to illustrate the possible value of T-DXd and chemotherapy at various WTP levels/QALY gains.

### Subgroup analyses

We conducted subgroup analyses to examine the variability in outcomes from different subgroups of patients with advanced HER2+ BC that were reported in the trials of DESTINY-Breast04 based on different PFS estimates. All statistical analyses were performed using R, version 4.0.5, 2021 (R Foundation for Statistical Computing), with the hesim and heemod packages.

## Results

### Base case analysis

A summary of all base-case results is presented in Table 2.

   **Scenario 1.**   In the base case analysis of the whole cohort of HER2-low advanced BC, T-DXd was associated with an increase in the effectiveness of 0.794 QALYs and 1.126 LYs, with an additional cost of $66,583 compared to chemotherapy. Accordingly, the ICER was $83,892/QALY. Furthermore, compared to chemotherapy, T-DXd had an INHB and INMB of 0.727 QALYs and $727,417 at a $100,000/QALY WTP threshold, respectively (Table 2).

   **Scenario 2.**   Compared to chemotherapy, T-DXd enhanced the effectiveness of 0.817 QALYs and 1.18 LYs in patients with HER2+ advanced BC, with an added cost of $67,602. In this case, the ICER was $82,802/QALY. Moreover, compared to chemotherapy, the INHB and INMB of T-DXd, at a $100,000/QALY WTP threshold, were 0.141 and $14,098, respectively (Table 2).

   **Scenario 3.**   Compared with chemotherapy, T-DXd led to an increase in the effectiveness of 0.737 QALYs and 1.226 LYs at an additional cost of $68,877. It is estimated that the ICER was $93,358/QALY. Additionally, T-DXd had an INHB of 0.048 QALYs and an INMB of $4,823 at a $100,000/QALY WTP threshold, compared to chemotherapy (Table 2).

### Sensitivity analysis

One-way sensitivity analyses suggested that the cost of T-DXd was the primary driver of the model outcome as well as its utility for PD and PFS in the three scenarios (S2 Fig). The key variables related to ICER were evaluated between T-DXd and chemotherapy in the three scenarios. For all HER2-Low advanced BC patients (Scenario 1), HER2+ advanced BC patients (Scenario 2), and HER2- advanced BC patients (Scenario 3), T-DXd was cost-effective at a WTP threshold of $50,000/QALY with a price of less than $17.00/mg, $17.13/mg, $14.07/mg, respectively (S3 Fig). The upper limit of the price range of T-DXd considered cost-effective at a WTP threshold of $100,000/QALY was $35.70/mg for all HER2-low advanced BC patients,

**Table 2. Summary of cost and outcome results in the base-case analysis.**

| Factor | Scenario 1 | | | Scenario 2 | | | Scenario 3 | | |
|---|---|---|---|---|---|---|---|---|---|
| | HER2-Low advanced BC patients | | | HER2+ advanced BC patients | | | HER2- advanced BC patients | | |
| | Trastuzumab deruxtecan | Chemotherapy | Incremental change | Trastuzumab deruxtecan | Chemotherapy | Incremental change | Trastuzumab deruxtecan | Chemotherapy | Incremental change |
| **Cost, $** | | | | | | | | | |
| Drug[a] | 62,971 | 19,507 | 43,464 | 63,311 | 19,783 | 43,528 | 60,819 | 17,911 | 42,908 |
| Nondrug[b] | 88,642 | 65,523 | 23,119 | 93,784 | 69,710 | 24,074 | 69,085 | 43,116 | 25,969 |
| Overall | 151,613 | 85,030 | 66,583 | 157,095 | 89,493 | 67,602 | 129,904 | 61,027 | 68,877 |
| **Life-years** | | | | | | | | | |
| Progression-free | 1.728 | 0.695 | 1.033 | 2.064 | 1.247 | 0.817 | 1.135 | 0.437 | 0.698 |
| Overall | 3.268 | 2.142 | 1.126 | 3.523 | 2.343 | 1.18 | 2.315 | 1.089 | 1.226 |
| **QALYs** | 1.945 | 1.151 | 0.794 | 2.064 | 1.247 | 0.817 | 1.371 | 0.634 | 0.737 |
| **ICERs, $** | | | | | | | | | |
| Per life-year | NA | NA | 59,129 | NA | NA | 57,309 | NA | NA | 56,217 |
| Per QALY | NA | NA | 83,892 | NA | NA | 82,802 | NA | NA | 93,358 |
| INHB, QALY, at threshold 100,000[a] | NA | NA | 0.727 | NA | NA | 0.141 | NA | NA | 0.048 |
| INMB, $, at threshold 100,000[a] | NA | NA | 727,417 | NA | NA | 14,098 | NA | NA | 4,823 |

Abbreviations: BC, breast cancer; HER2, human epidermal growth factor receptor 2; HER2+, human epidermal growth factor receptor 2 positive; HER2-, human epidermal growth factor receptor 2 negative; ICER, incremental cost-effectiveness ratio; INHB, incremental net health benefit; INMB, incremental net monetary benefit; NA, not applicable; QALYs, quality-adjusted life years.

[a]Compared with chemotherapy.

[b]Nondrug cost includes the costs of adverse event management, subsequent best supportive care per patient, and follow-up care covering physician monitors, drug administration, and terminal care.

$35.27/mg for all HER2+ advanced BC patients and $32.07/mg for all HER2- advanced BC patients.

The cost-effectiveness acceptability curves provide a visual representation of the results of the probabilistic sensitivity analysis (Fig 2). As the WTP threshold increased, there was a greater likelihood that T-DXd would be cost-effective. In comparison to chemotherapy, the probabilities of T-DXd being considered cost-effective were 81.10%, 82.27%, and 73.78%, respectively, based on a WTP threshold of $100,000/QALY for patients with HER2-low advanced BC patients, HER2+ advanced BC patients, and HER2- advanced BC patients.

### Subgroup analysis

A subgroup analysis was performed by varying the hazard ratios (HRs) of PFS. Based on the subgroup analysis, T-DXd was associated with lower HRs than chemotherapy in most of the subgroups; therefore, at a WTP of $100,000/QALY, T-DXd was > 70%. Hence, T-DXd may be considered cost-effective (Table 3).

### Discussion

In this study, we conducted a cost-effectiveness analysis of T-DXd vs chemotherapy for previously treated HER2-low advanced BC, and the results revealed that compared with chemotherapy, the ICERs of T-DXd was $83,892/QALY, $82,808/QALY, and $93,358/QALY in all

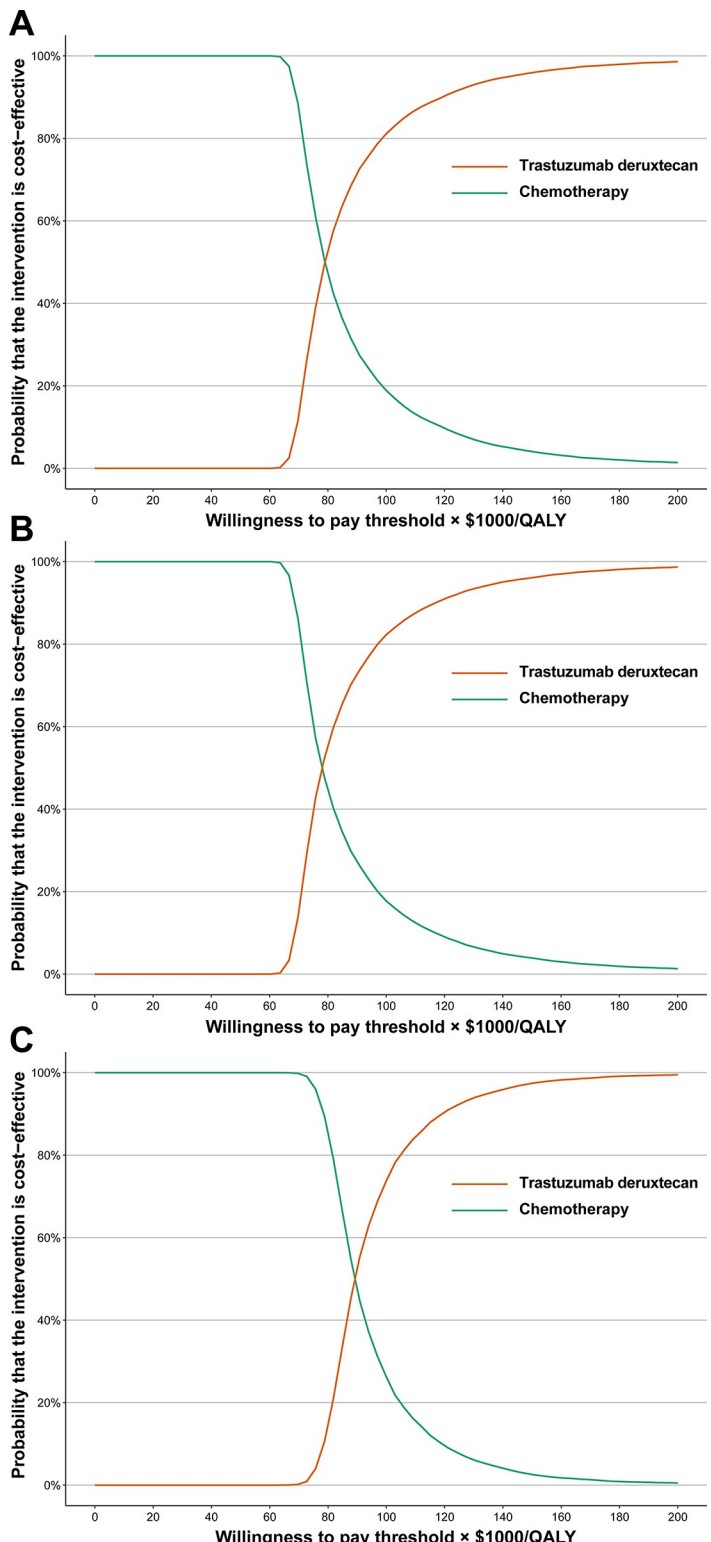

**Fig 2. Acceptability curves of cost-effectiveness for trastuzumab deruxtecan versus chemotherapy.** (A) All HER2-low advanced BC patients, (B) HER2+ advanced BC patients, (C) HER2- advanced BC patients.

**Table 3. Summary of subgroup analyses obtained by varying the HRs for PFS.**

| Subgroup | Unstratified HR for PFS (95% CI)[a] | Change in cost, $[b] | Change in QALY[b] | ICER, $/QALY | T-DXd vs. chemotherapy at WTP threshold of $100,000/QALY | |
|---|---|---|---|---|---|---|
| | | | | | Cost-effectiveness probability of T-DXd, % | INHB |
| **Prior CDK 4/6 inhibitors** | | | | | | |
| Yes | 0.55 (0.42, 0.73) | 66,992 | 0.760 | 88,100 | 78.88 | 0.090 |
| No | 0.42 (0.28, 0.64) | 69,043 | 0.978 | 70,608 | 88.11 | 0.287 |
| **IHC status** | | | | | | |
| IHC 1+ | 0.48 (0.35, 0.64) | 68,071 | 0.864 | 78,766 | 84.64 | 0.184 |
| IHC 2+/ISH- | 0.55 (0.38, 0.80) | 66,992 | 0.760 | 88,100 | 78.49 | 0.090 |
| **Prior lines of chemotherapy in the metastatic setting** | | | | | | |
| 1 | 0.54 (0.40, 0.73) | 67,143 | 0.774 | 86,786 | 79.63 | 0.102 |
| > 2 | 0.47 (0.33, 0.68) | 68,229 | 0.881 | 77,412 | 85.74 | 0.199 |
| **Age** | | | | | | |
| < 65 | 0.51 (0.39, 0.67) | 67,586 | 0.815 | 82,802 | 82.54 | 0.140 |
| ≥ 65 | 0.47 (0.29, 0.77) | 68,229 | 0.881 | 77,412 | 85.04 | 0.199 |
| **Race** | | | | | | |
| White | 0.64 (0.44, 0.91) | 65,678 | 0.660 | 99,510 | 57.74 | 0.003 |
| Asian | 0.40 (0.28, 0.56) | 69,379 | 1.022 | 67,884 | 88.97 | 0.328 |
| Other | 0.83 (0.41, 1.69) | 63,112 | 0.524 | 120,356 | 8.49 | -0.107 |
| **Region** | | | | | | |
| Asian | 0.41 (0.28, 0.58) | 69,210 | 0.999 | 69,245 | 88.02 | 0.307 |
| Europe and Israel | 0.62 (0.43, 0.89) | 65,964 | 0.680 | 97,047 | 65.15 | 0.020 |
| North America | 0.54 (0.30, 0.97) | 67,143 | 0.774 | 86,786 | 79.71 | 0.102 |
| **ECOG performance status** | | | | | | |
| 0 | 0.56 (0.40, 0.77) | 66,842 | 0.748 | 89,406 | 76.43 | 0.079 |
| 1 | 0.45 (0.32, 0.64) | 68,551 | 0.918 | 74,695 | 87.03 | 0.232 |
| **Visceral disease at baseline** | | | | | | |
| Yes | 0.54 (0.42, 0.69) | 67,143 | 0.774 | 86,786 | 79.42 | 0.102 |
| No | 0.23 (0.09, 0.55) | 70,789 | 1.239 | 57,122 | 90.62 | 0.531 |

Abbreviations: CDK, cyclin-dependent kinase; ECOG, Eastern Cooperative Oncology Group; HR, hazard ratio; ICER, incremental cost-effectiveness ratio; IHC, immunohistochemical, INHB, incremental net health benefits; PFS, progression-free survival; QALY, quality-adjusted life-year; WTP, willingness to pay.

[a]HR for PFS represents the HR of T-DXd vs. chemotherapy for PFS

[b]Change in cost and change in QALYs represent the results of T-DXd minus chemotherapy.

HER2-Low advanced BC patients (Scenario 1), HER2+ advanced BC patients (Scenario 2) and HER2- advanced BC patients (Scenario 3), respectively. Even the highest ICER value remains below the threshold for WTP of $100,000/QALY. Based on the results of this analysis, it was apparent that the HER2+ group had a higher survival rate, which was related to a lower ICER. The results of this model were robust, as determined by comprehensive deterministic and probabilistic sensitivity analyses. The results of the one-way sensitivity analysis suggested that T-DXd cost, HR for PFS, and OS played a significant role in determining sensitivity. T-DXd would be cost-effective if its price were reduced to $17.00/mg, $17.13/mg, and $14.07/mg at a WTP threshold of $50,000/QALY for all HER2-low advanced BC and HER2-advanced BC, respectively. The probabilistic sensitivity analysis of T-DXd revealed that for all advanced BC patients with HER2-Low, HER2+, and HER2-status, the probability of cost-effectiveness of T-DXd was 81.10%, 82.27%, and 73.78%, respectively, at $100,000/QALY. In patients with

previously treated HER2-low advanced BC, T-DXd treatment is a cost-effective alternative to chemotherapy.

This cost-effectiveness analysis for T-DXd in treating HER2-low advanced BC is a novel approach that considers both parametric and structural uncertainties. We believe that this is the first study to examine the cost-effectiveness of T-DXd compared to chemotherapy for patients with previously treated advanced HER2-low BC. Cancer treatment is often limited by the fact that the most clinically effective drugs are not the most cost-effective. By establishing prices for novel treatments based on the value of the treatment they provide, value-based pricing is a new method of setting prices for novel treatments that are logically based on the fact that the optimal oncology treatment is becoming increasingly less cost-effective [37]. Therefore, it is important to discuss the clinical outcomes of the study. The effectiveness of T-DXd in the treatment of metastatic BC associated with HER2+ status was previously compared with that of trastuzumab emtansine. Despite its clinical benefits in the treatment of HER2+ metastatic BC, T-DXd is not cost-effective because of its high cost [38]. Nevertheless, in this study, we conclude that T-DXd may be a cost-effective alternative to chemotherapy in patients with previously treated HER2-low advanced BC.

The results of the one-way sensitivity analysis suggested that T-DXd cost was a sensitive variable. The price of T-DXd remains the most sensitive variable, and reducing the price of T-DXd is important to increase its feasibility. On August 5, 2022, the Food and Drug Administration approved trastuzumab deruxtecan for adult patients with HER2-low advanced BC who had received prior chemotherapy in a metastatic setting or developed disease recurrence during or within six months of completing adjuvant chemotherapy. However, T-DXd is not included in Medicare Part B in the United States. If Medicare Part B covers T-DXd and patients pay the Medicare Part B deductible, Medicare beneficiaries will have a certain percent co-insurance cost for T-DXd. The government announced American Patients First and aimed to construct a blueprint for cutting drug prices and reducing out-of-pocket payments in the United States [39, 40]. The availability of innovative treatments requires significant reductions in price and financial assistance.

It is important to note the advantages of this study. First, to our knowledge, this is the first study comparing T-DXd treatment with chemotherapy in patients with previously treated advanced HER2-low BC from the perspective of a third-party payer in the United States. The partitioned survival model was adopted to evaluate cost-effectiveness analysis. Second, T-DXd treatment may be a cost-effective alternative to chemotherapy in patients with previously treated HER2-low advanced BC. Third, physicians and patients may benefit from economic information regarding the expression levels of HER2 and other subgroups to make more informed treatment decisions.

Some limitations of the study warrant further consideration. First, health outcomes that extended beyond the duration of the DESTINY-Breast04 trial were assumed by fitting parametric distributions to the K-M OS and PFS data, which may have resulted in inaccurate results. Based on the results of the sensitivity analysis, this limitation may not be a significant factor; accordingly, this conclusion is generally considered robust. Second, the clinical parameters in the model were based on the results of the DESTINY-Breast04 clinical trial. Thus, biases within the trial may have affected the cost and effectiveness of the results. Clinical trial participants had a higher rate of adherence to treatment regimens than those participating in real-world settings. Third, it is important to note that the utilities used in the model were not estimated using DESTINY-Breast04, but rather from other health utility surveys in patients with advanced HER2-low BC. The cost-effectiveness analysis is also influenced by the assumption that patients in both groups have the same utility, which may lead to some bias in the results. Last, the grades 1 and 2 AEs were not taken into account. The findings in the one-way

sensitivity analyses suggest that these limitations may not have a significant impact, as the costs and disutilities associated with AEs were relatively minor. In clinical practice, these adverse events cannot be ignored.

## Conclusion

The results of these estimates indicate that from the perspective of third-party payers in the United States, T-DXd treatment may be a cost-effective alternative to chemotherapy for patients with prior treatment for HER2-low advanced BC. Economic outcomes can be improved by tailoring treatments to patients' needs. Clinicians may use these results to select appropriate treatment options for patients with HER2-low advanced BC.

## Supporting information

**S1 Fig. Model fitting analysis.**
(PDF)

**S2 Fig. Tornado diagram of 1-way sensitivity analyses of pembrolizumab versus chemo-therapy.**
(PDF)

**S3 Fig. One-way sensitivity analyses result of ICER when varying cost of trastuzumab der-uxtecan in participants.**
(PDF)

**S1 Table. Estimated parameters and AIC and BIC values from each survival model.**
(PDF)

**S2 Table. Associated costs and disutility of grade $\geq$ 3 treatment-related adverse events.**
(PDF)

## Author Contributions

**Conceptualization:** Demin Shi, Xueyan Liang, Lingyuan Chen.

**Data curation:** Demin Shi, Yan Li.

**Formal analysis:** Demin Shi, Xueyan Liang, Yan Li.

**Funding acquisition:** Lingyuan Chen.

**Methodology:** Demin Shi, Xueyan Liang, Yan Li.

**Project administration:** Xueyan Liang, Lingyuan Chen.

**Software:** Yan Li.

**Supervision:** Yan Li, Lingyuan Chen.

**Validation:** Xueyan Liang.

**Visualization:** Xueyan Liang.

**Writing – original draft:** Demin Shi, Xueyan Liang.

**Writing – review & editing:** Xueyan Liang, Yan Li, Lingyuan Chen.

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
