## [Decision Letter · Decision Letter 0]

13 Jun 2023

PONE-D-23-04529Cost-effectiveness of Trastuzumab Deruxtecanfor Previously Treated HER2-Low Advanced Breast CancerPLOS ONE

Dear Dr. chen,

Thank you for submitting your manuscript to PLOS ONE. After careful consideration, we feel that it has merit but does not fully meet PLOS ONE’s publication criteria as it currently stands. Therefore, we invite you to submit a revised version of the manuscript that addresses the points raised during the review process.

We look forward to receiving your revised manuscript.

Kind regards,

Mirosława Püsküllüoğlu, MD, PhD

Academic Editor

PLOS ONE

5. Please amend either the title on the online submission form (via Edit Submission) or the title in the manuscript so that they are identical.

Reviewers' comments:

Reviewer's Responses to Questions

**Comments to the Author**

1. Is the manuscript technically sound, and do the data support the conclusions?

Reviewer #1: Yes

Reviewer #2: Yes

2. Has the statistical analysis been performed appropriately and rigorously? 

Reviewer #1: Yes

Reviewer #2: Yes

3. Have the authors made all data underlying the findings in their manuscript fully available?

Reviewer #1: Yes

Reviewer #2: Yes

4. Is the manuscript presented in an intelligible fashion and written in standard English?

Reviewer #1: Yes

Reviewer #2: Yes

5. Review Comments to the Author

Reviewer #1: This study evaluated the cost-effectiveness of trastuzumab deruxtecan (T-DXd) among HER2-Low advanced breast cancer (BC) patients, and the results showed that T-DXd is a cost-effective alternative to chemotherapy at willingness to pay threshold of $100,000/QALY. This discovery is intriguing and could have significant implications for the patients, clinicians and third-party payers. The methods are appropriate and results are reliable. However, I have several minor comments.

1. For subgroup analyses, why did you only focus on PFS, how about OS? How did the PFS variate?

2. For one-way sensitivity analyses among HER2- advanced BC patients (Supplementary Figure S2-C), the results showed that variation of the top 5 parameters might result in the ICER being over $100,000/QALY, which indicated that T-DXd was not cost-effective. How do we understand this result since you mentioned the results of this model are robust?

3. I did not find Figure 2 in the manuscript.

4. For the result of subgroup analysis, the manuscript stated “T-DXd was associated with lower HRs than chemotherapy in most of the subgroups”, how can conclude it from Table 3?

5. Is the T-DXd sole drug that is cost-effective for HER2-Low advanced BC patients? If not, what are the advantages of T-DXd?

6. I suggested that the coverage information of T-DXd in public healthcare systems such as Medicare should be added.

Reviewer #2: The manuscript is technically sound. Only a few minor suggestions:

Please clearly highlight the limitations of the study in the discussion section with some justifications

There are some typo and grammatical errors so a proof reading is advised.

Overall the paper is structured well.

6. PLOS authors have the option to publish the peer review history of their article (what does this mean?). If published, this will include your full peer review and any attached files.

Reviewer #1: No

Reviewer #2: No

---

## [Author Response · Author response to Decision Letter 0]

7 Jul 2023

Dear Reviewers and Editors,

Thank you for your extremely efficient and earnest work on our manuscript. The comments are all valuable and very helpful for revising and improving our paper. The main corrections in the manuscript and the response to the Referees' comments are listed below point by point:

Reviewer(s)' Comments:

Reviewer #1: This study evaluated the cost-effectiveness of trastuzumab deruxtecan (T-DXd) among HER2-Low advanced breast cancer (BC) patients, and the results showed that T-DXd is a cost-effective alternative to chemotherapy at willingness to pay threshold of $100,000/QALY. This discovery is intriguing and could have significant implications for the patients, clinicians and third-party payers. The methods are appropriate and results are reliable. However, I have several minor comments.

1. For subgroup analyses, why did you only focus on PFS, how about OS? How did the PFS variate?

Reply: We conducted subgroup analyses to examine the variability in outcomes from different subgroups of patients with advanced HER2+ BC that were reported in the trials of DESTINY-Breast04. Due to the subgroup results of OS was not provided, we did not perform the subgroup analysis focus on OS. In the DESTINY-Breast04 trial, a consistent benefit was observed for trastuzumab deruxtecan across analyzed subgroups, trastuzumab deruxtecan showed superior activity over standard chemotherapy options in patients with HER2-low advanced breast cancer in most subgroups except for other race. Furthermore, the unstratified HR for PFS was shown in Table 3. For most subgroups in this study, the ICERs were greatly influenced by HRs, and trastuzumab deruxtecan performed better when the risks of death were lower.

2. For one-way sensitivity analyses among HER2- advanced BC patients (Supplementary Figure S2-C), the results showed that variation of the top 5 parameters might result in the ICER being over $100,000/QALY, which indicated that T-DXd was not cost-effective. How do we understand this result since you mentioned the results of this model are robust?

Reply: Due to we considered a wider range of the base-case values (± 20%) for the parameters and ICER being very close to $100,000/QALY, the results showed that variation of the top 5 parameters might result in the ICER being over $100,000/QALY.

In all cases, a suitable distribution was determined based on different parameters. In terms of cost parameters, proportion and probability, and baseline characteristics, gamma, beta, and normal distributions were assigned, respectively. We selected the justified distribution and sensitivity analysis undertaken to assess the robustness of estimates to this choice. In the past, the most common form of sensitivity analysis was to undertake a one-way analysis. Here estimates for each parameter are varied one at a time in order to investigate the impact on study results. A common way to present the results of a one-way analysis is in a ‘tornado diagram’. The impact that variation in each parameter has on the study result is shown by the width of the respective band. These diagrams are normally arranged so that the parameter in which variation has the biggest impact on the study result is at the top. A variant of one-way sensitivity analysis is to undertake a threshold analysis. Here the critical value(s) of a parameter or parameters central to the decision are identified. For example, a decision-maker might specify an increase in cost, or an ICER, above which the program would not be acceptable. Then the analyst could assess which combinations of parameter estimates could cause the threshold to be exceeded. Alternatively, the threshold values for key parameters that would cause the program to be too costly or not cost-effective could be defined. The decision-makers could then make a judgment about whether particular thresholds were likely to be breached or not. 

Hence, the conclusion of this study is that the result is robust.

3. I did not find Figure 2 in the manuscript.

Reply: We have reuploaded Figure 2.

4. For the result of subgroup analysis, the manuscript stated “T-DXd was associated with lower HRs than chemotherapy in most of the subgroups”, how can conclude it from Table 3?

Reply: The results were illustrated in Table 3, and the unstratified HR for PFS (95% CI) showed the results that T-DXd was associated with lower HRs than chemotherapy in most of the subgroups.

5. Is the T-DXd sole drug that is cost-effective for HER2-Low advanced BC patients? If not, what are the advantages of T-DXd?

Reply: The aim of this study was to evaluate the cost-effectiveness of T-DXd vs. chemotherapy for previously treated HER2-low advanced BC, and the comparison of this study was T-DXd and chemotherapy, and no other drugs were evaluated. From a third-party payer's perspective in the United States, the findings of the cost-effectiveness analysis revealed that, at the current price, T-DXd is a cost-effective alternative to chemotherapy for patients with prior HER2-low advanced BC, at WTP threshold of $100,000/QALY, and this is the advantage of T-DXd.

6. I suggested that the coverage information of T-DXd in public healthcare systems such as Medicare should be added.

Reply: Thank you for your suggestion. We have added that information in the section of discussion.

The results of the one-way sensitivity analysis suggested that T-DXd cost was a sensitive variable. The price of T-DXd remains the most sensitive variable, and reducing the price of T-DXd is important to increase its feasibility. On August 5, 2022, the Food and Drug Administration approved trastuzumab deruxtecan for adult patients with HER2-low advanced BC who had received prior chemotherapy in a metastatic setting or developed disease recurrence during or within six months of completing adjuvant chemotherapy. However, T-DXd is not included in Medicare Part B in the United States. If Medicare Part B covers T-DXd and patients pay the Medicare Part B deductible, Medicare beneficiaries will have a certain percent co-insurance cost for T-DXd. The government announced American Patients First and aimed to construct a blueprint for cutting drug prices and reducing out-of-pocket payments in the United States [39, 40]. The availability of innovative treatments requires significant reductions in price and financial assistance.

Reviewer #2: The manuscript is technically sound. Only a few minor suggestions:

Please clearly highlight the limitations of the study in the discussion section with some justifications

There are some typo and grammatical errors so a proof reading is advised.

Overall the paper is structured well.

Reply: Thank you for your suggestion.

We have provided and highlight the limitations of the study in the discussion section with justifications.

We have corrected the typo and grammatical errors in our manuscript and English has been edited and proofread by editage.

---

## [Editor Report · Decision Letter 1]

10 Aug 2023

Cost-effectiveness of Trastuzumab Deruxtecan for Previously Treated HER2-Low Advanced Breast Cancer

PONE-D-23-04529R1

Dear Dr. Chen

We’re pleased to inform you that your manuscript has been judged scientifically suitable for publication and will be formally accepted for publication once it meets all outstanding technical requirements.

Kind regards,

Mirosława Püsküllüoğlu, MD, PhD

Academic Editor

PLOS ONE
---

## [Editor Report · Acceptance letter]

15 Aug 2023

PONE-D-23-04529R1 

Cost-effectiveness of Trastuzumab Deruxtecan for Previously Treated HER2-Low Advanced Breast Cancer 

Dear Dr. chen:

I'm pleased to inform you that your manuscript has been deemed suitable for publication in PLOS ONE. Congratulations! Your manuscript is now with our production department. 

Kind regards, 

on behalf of

Dr. Mirosława Püsküllüoğlu 

Academic Editor

PLOS ONE